# Levels of Pro-Inflammatory and Bone-Resorptive Mediators in Periodontally Compromised Patients under Orthodontic Treatment Involving Intermittent Forces of Low Intensities

**DOI:** 10.3390/ijms24054807

**Published:** 2023-03-02

**Authors:** Cristian Navarrete, Alejandro Riquelme, Natalia Baksai, Romina Pérez, Claudia González, María Michea, Hans von Mühlenbrock, Emilio A. Cafferata, Rolando Vernal

**Affiliations:** 1Orthodontics-Periodontics Center, Faculty of Dentistry, Universidad de Chile, Santiago 8380492, Chile; 2Department of Periodontology, School of Dentistry, Universidad Científica del Sur, Lima 15067, Peru; 3Periodontal Biology Laboratory, Faculty of Dentistry, Universidad de Chile, Santiago 8380492, Chile; 4Department of Conservative Dentistry, Faculty of Dentistry, Universidad de Chile, Santiago 8380492, Chile

**Keywords:** periodontally compromised teeth, reduced periodontal support, low-intensity orthodontic forces, RANKL/OPG ratio, interleukin-6, interleukin-17A, matrix metalloproteinase-8

## Abstract

During orthodontic treatment, diverse cytokines, enzymes, and osteolytic mediators produced within the teeth surrounding periodontal tissues determine the rate of alveolar bone remodeling and consequent teeth movement. In patients with teeth presenting reduced periodontal support, periodontal stability should be ensured during orthodontic treatment. Thus, therapies based on the application of low-intensity intermittent orthodontic forces are recommended. To determine if this kind of treatment is periodontally well tolerated, this study aimed to analyze the production of receptor activator of nuclear factor kappa-B ligand (RANKL), osteoprotegerin (OPG), interleukin (IL)-6, IL-17A, and matrix metalloproteinase (MMP)-8 in periodontal tissues of protruded anterior teeth with reduced periodontal support and undergoing orthodontic treatment. Patients with periodontitis-associated anterior teeth migration received non-surgical periodontal therapy and a specific orthodontic treatment involving controlled low-intensity intermittent orthodontic forces. Samples were collected before periodontitis treatment, after periodontitis treatment, and at 1 week to 24 months of the orthodontic treatment. During the 2 years of orthodontic treatment, no significant differences were detected in the probing depth, clinical attachment level, supragingival bacterial plaque, and bleeding on probing. In line with this, the gingival crevicular levels of RANKL, OPG, IL-6, IL-17A, and MMP-8 did not vary between the different evaluation time-points of the orthodontic treatment. When compared with the levels detected during the periodontitis, the RANKL/OPG ratio was significantly lower at all the analyzed time-points of the orthodontic treatment. In conclusion, the patient-specific orthodontic treatment based on intermittent orthodontic forces of low intensities was well tolerated by periodontally compromised teeth with pathological migration.

## 1. Introduction

A frequent reason for patients seeking dental treatment is the pathological migration of anterior teeth due to periodontitis-associated reduced periodontal support [1]. After periodontal treatment, anterior teeth may exhibit reduced periodontal support as a consequence of periodontitis and, in response to the various physiological forces they are subjected to, frequently undergo pathological migration [2,3]. In these periodontally compromised teeth, it has been proposed that successful orthodontic treatment would improve function and esthetics [4,5]. Indeed, orthodontic treatment would not cause additional periodontal damage if controlled low-intensity orthodontic forces are used and the patient achieves a suitable oral biofilm control [2,4,6].

Quantitative and qualitative changes in the oral biofilm are frequently observed in orthodontic patients [7,8]. Indeed, significant changes in the amount and composition of dental plaque occur as early as the first week and become more consistent from the third month after the start of orthodontic treatment [9]. Different orthodontic appliances tend to alter the oral microbiota during treatment, and fixed appliances have been shown to have a more significant accumulation of specific bacteria associated with gingivitis and periodontitis etiology, including *Porphyromonas gingivalis*, *Prevotella intermedia*, *Tannerella forsythia*, and *Aggregatibacter actinomycetemcomitans*, as compared with removable appliances [10]. It is noteworthy that these microbiological changes do not affect periodontal parameters if adequate follow-up with the patient is carried out and efficient control of dental plaque is achieved [7,8]. Thus, the strengthening of oral hygiene and periodical periodontal controls during the entire orthodontic treatment are recommended.

Apart from that, the analysis of gingival crevicular fluid (GCF) samples has reported an increase in the levels of pro-inflammatory cytokines, matrix metalloproteinases (MMPs), and extracellular matrix components at the tooth surface towards which orthodontic movement was directed, a consequence of the biological changes in the periodontal soft tissues around the teeth undergoing movement [11,12,13]. Similarly, an increase in the levels of the bone resorptive mediator called receptor activator of nuclear factor kappa-B ligand (RANKL), as well as a decrease in the levels of the bone protective mediator called osteoprotegerin (OPG), have been detected in the periodontal compression sites during conventional orthodontic movements, which are directly associated with orthodontically induced bone remodeling [14]. Interestingly, when controlled low-intensity orthodontic forces were applied on teeth with reduced periodontal support, no significant changes in the GCF levels of diverse MMPs were detected, suggesting an absence of the deleterious changes in the periodontium as a consequence of this orthodontic treatment, at least in the soft tissues [15].

With the goal of clarifying several aspects of the biological changes that occur in the periodontium of teeth with reduced periodontal support undergoing orthodontic treatment, the aim of this study was to quantify the changes in the GCF levels of specific biomarkers of inflammation and alveolar bone resorption. We hypothesized that controlled and intermittent low-intensity orthodontic forces are well tolerated by periodontally compromised anterior teeth with pathological migration. In particular, we hypothesized that the GCF levels of RANKL, OPG, IL-6, IL-17A, and MMP-8 do not change during this orthodontic treatment and are lower than those detected during periodontitis.

## 2. Results

### 2.1. Study Patients and Clinical Evaluation

This study included 42 patients affected with periodontitis and presenting protruding anterior teeth. These patients were periodontally treated and after 6 months of supportive periodontal therapy, they were orthodontically treated using controlled and intermittent low-intensity orthodontic forces. During the 2 years of orthodontic treatment, no significant differences were detected in the probing depth (PD), clinical attachment level (CAL), supragingival bacterial plaque (BP), and bleeding on probing (BOP), clinical parameters used to perform the periodontitis diagnosis, and post-treatment periodontal follow-up (Table 1).

### 2.2. Molecular Evaluation

In the anterior protruded teeth, the secreted levels of RANKL, OPG, IL-6, IL-17A, and MMP-8 were analyzed in GCF samples obtained from four different periodontal sites: vestibular, palatine, mesial, and distal. Figure 1, Figure 2, Figure 3, Figure 4, Figure 5 and Figure 6 show the concentration of the different molecular mediators at the different evaluation times: before starting periodontal treatment (periodontitis), after the end of the supportive periodontal therapy (post-periodontitis), and at seven time-points during the orthodontic treatment. In each figure, panel “a” shows the total concentration of each molecular mediator, corresponding to all the sampled periodontal sites in each tooth. Instead, panel “b” shows the concentration of each molecular mediator per periodontal site. Thus, panel “a” allows the comparison of the molecular levels between the different evaluation time-points of the orthodontic treatment and when the patient was affected with periodontitis. In contrast, panel “b” allows the comparison of the variations in the molecular mediators between the four periodontal sites. Particularly, to compare against the buccal periodontal site, where the tooth had pathologically migrated, and against the palatal periodontal site, where the main orthodontic force was applied.

### 2.3. RANKL Levels

Due to the fact that orthodontic-induced teeth movements depend on changes in the alveolar bone metabolism and moreover, RANKL and OPG regulate osteoclast differentiation and bone resorption, herein, we analyzed their levels in the GCF samples.

The total RANKL levels detected in periodontally compromised teeth did not significantly vary between the different time-points of evaluation for the orthodontic treatment (Figure 1a). During periodontitis, the RANKL levels detected in the vestibular periodontal site (331.64 pg/mL) were significantly higher than those detected in the palatine (240.88 pg/mL, *p* = 0.017), distal (249.21 pg/mL, *p* = 0.025), and mesial (251.76 pg/mL, *p* = 0.031) sites (Figure 1b). During the orthodontic treatment, the vestibular RANKL levels decreased and the palatine RANKL levels increased, in concordance with the applied orthodontic forces; however, these increased palatine levels were not statistically different from those detected in the other periodontal sites (Figure 1b).

**Figure 1 ijms-24-04807-f001:**
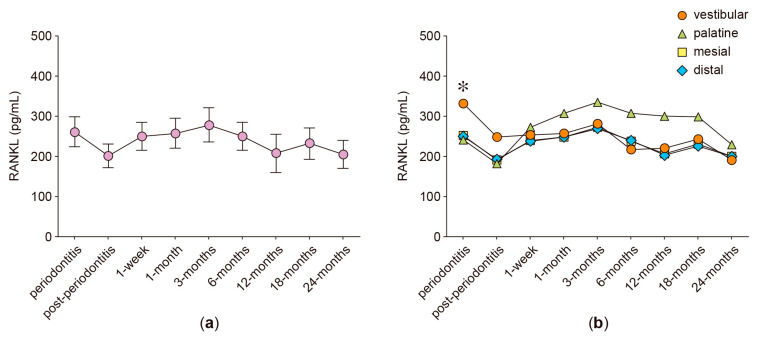
RANKL levels. (**a**) RANKL levels detected in the gingival crevicular fluid of periodontally compromised anterior teeth before periodontitis treatment (periodontitis), after periodontitis treatment (post-periodontitis), and during 2 years of patient-specific orthodontic treatment (1 week to 24 months), using intermittent low-intensity orthodontic forces. Data are represented as pg/mL and are shown as mean ± SD. (**b**) RANKL levels detected in each periodontal site, analyzed at the same time-points described in (**a**). * *p* < 0.05 at periodontitis examination, comparing the vestibular site with the other periodontal sites.

### 2.4. OPG Levels

After periodontitis treatment, the total OPG levels increased (82.66 pg/mL) when compared with those detected at the periodontitis examination (47.49 pg/mL, *p* = 0.005) and remained increased during the entire orthodontic treatment: 1 week (88.60 pg/mL, *p* = 0.001), 1 month (91.32 pg/mL, *p* < 0.001), 3 months (93.35 pg/mL, *p* < 0.001), 6 months (104.02 pg/mL, *p* = 0.004), 12 months (98.29 pg/mL, *p* = 0.008), 18 months (95.91 pg/mL, *p* = 0.002), and 24 months (94.81 pg/mL, *p* = 0.008) (Figure 2a). This OPG increment was also detected when the different periodontal sites were separately analyzed (Figure 2b).

**Figure 2 ijms-24-04807-f002:**
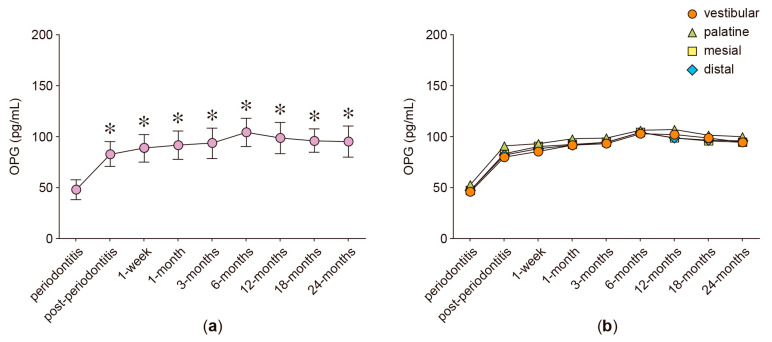
OPG levels. (**a**) OPG levels detected in the gingival crevicular fluid of periodontally compromised anterior teeth before periodontitis treatment (periodontitis), after periodontitis treatment (post-periodontitis), and during 2 years of patient-specific orthodontic treatment (1 week to 24 months), using intermittent low-intensity orthodontic forces. Data are represented as pg/mL and shown as mean ± SD. (**b**) OPG levels detected in each periodontal site, analyzed at the same time-points described in (**a**). * *p* < 0.05 periodontitis compared with post-periodontitis and all the time-points of orthodontic treatment.

### 2.5. RANKL/OPG Ratio

Consistently, after the periodontitis treatment, the RANKL/OPG ratio significantly diminished (post-periodontitis ratio 2.65) when compared with the RANKL/OPG ratio detected before the periodontitis treatment (periodontitis ratio 6.67, *p* < 0.001) (Figure 3a). During the orthodontic treatment, the RANKL/OPG ratio did not vary when compared with the ratio detected at the post-periodontitis examination; however, this RANKL/OPG ratio was lower at all the analyzed time-points when compared with the RANKL/OPG ratio detected during periodontitis: 1 week (ratio 3.14, *p* < 0.001), 1 month (ratio 3.06, *p* < 0.001), 3 months (ratio 3.20, *p* < 0.001), 6 months (ratio 2.44, *p* = 0.004), 12 months (ratio 2.15, *p* = 0.001), 18 months (ratio 2.53, *p* < 0.001), and 24 months (ratio 2.34, *p* = 0.001) (Figure 3a). This decreased RANKL/OPG ratio was also found when the different periodontal sites were separately analyzed (Figure 3b).

**Figure 3 ijms-24-04807-f003:**
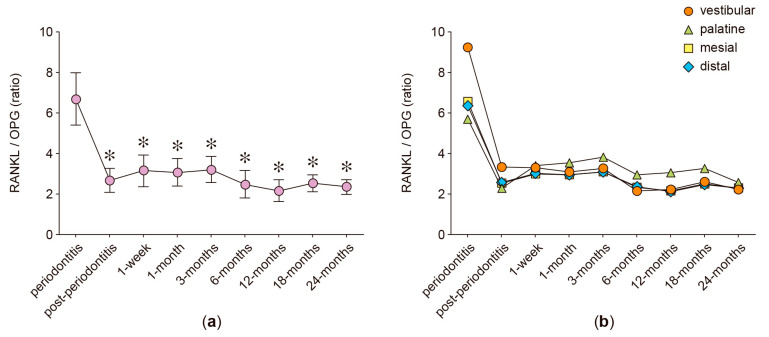
RANKL/OPG ratio. (**a**) RANKL/OPG ratio detected in the gingival crevicular fluid of periodontally compromised anterior teeth before periodontitis treatment (periodontitis), after periodontitis treatment (post-periodontitis), and during 2 years of patient-specific orthodontic treatment (1 week to 24 months), using intermittent low-intensity orthodontic forces. Data are represented as ratios and shown as mean ± SD. (**b**) RANKL/OPG ratio detected in each periodontal site, analyzed at the same time-points described in (**a**). * *p* < 0.05 periodontitis compared with post-periodontitis and all the time-points of orthodontic treatment.

### 2.6. IL-6 Levels

During the orthodontic-induced teeth movement, RANKL is mainly produced by periodontal fibroblasts and T-helper type-17 (Th17) lymphocytes. Th17-type cytokines are also implied in periodontal inflammation; thus, we also analyzed the IL-6 and IL-17A production in the GCF samples obtained from the periodontally compromised anterior teeth under orthodontic treatment.

The secreted levels of IL-6 did not significantly vary between the different evaluation time-points of the orthodontic treatment (Figure 4a). Additionally, no significant differences were detected when the IL-6 levels were analyzed separately and between the different periodontal sites (Figure 4b).

**Figure 4 ijms-24-04807-f004:**
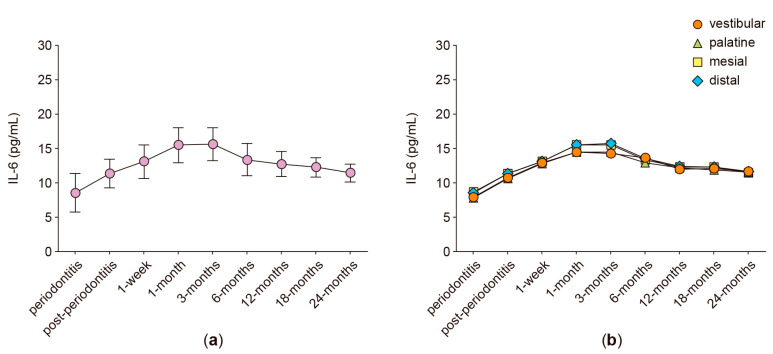
IL-6 levels. (**a**) IL-6 levels detected in the gingival crevicular fluid of periodontally compromised anterior teeth before periodontitis treatment (periodontitis), after periodontitis treatment (post-periodontitis), and during 2 years of patient-specific orthodontic treatment (1 week to 24 months), using intermittent low-intensity orthodontic forces. Data are represented as pg/mL and shown as mean ± SD. (**b**) IL-6 levels detected in each periodontal site, analyzed at the same time-points described in (**a**).

### 2.7. IL-17A Levels

Similarly, the secreted levels of IL-17A did not significantly vary between the different evaluation time-points of the orthodontic treatment (Figure 5a). Moreover, no significant differences were detected when the IL-17A levels were analyzed separately and between the different periodontal sites (Figure 5b).

**Figure 5 ijms-24-04807-f005:**
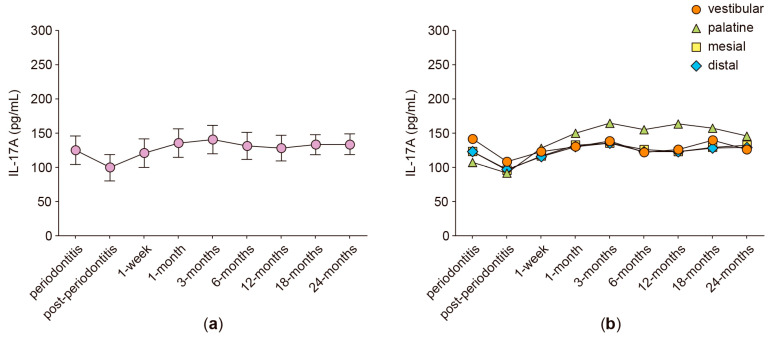
IL-17A levels. (**a**) IL-17A levels detected in the gingival crevicular fluid of periodontally compromised anterior teeth before periodontitis treatment (periodontitis), after periodontitis treatment (post-periodontitis), and during 2 years of patient-specific orthodontic treatment (1 week to 24 months), using intermittent low-intensity orthodontic forces. Data are represented as pg/mL and shown as mean ± SD. (**b**) IL-17A levels detected in each periodontal site, analyzed at the same time-points described in (**a**).

### 2.8. MMP-8 Levels

Finally, the MMP-8 produced levels were also analyzed because it is the more directly implied MMP in the course of collagen degradation within the soft periodontal tissues during the orthodontic movements.

Lower levels of MMP-8 were detected during the examination carried out after the periodontitis treatment (1912.19 pg/mL) compared with the examination previous to the periodontitis treatment (2272.08 pg/mL, *p* = 0.001) (Figure 6a). During the orthodontic treatment, the MMP-8 levels did not vary when compared with those detected at the post-periodontitis examination. However, the MMP-8 levels were lower at all the analyzed time-points of the orthodontic treatment when compared with those detected during periodontitis: 1 week (1970.41 pg/mL, *p* = 0.011), 1 month (2013.12 pg/mL, *p* < 0.040), 3 months (1996.23 pg/mL, *p* = 0.045), 6 months (1951.11 pg/mL, *p* = 0.042), 12 months (1887.88 pg/mL, *p* = 0.047), 18 months (1828.26 pg/mL, *p* = 0.001), and 24 months (1675.53 pg/mL, *p* < 0.001) (Figure 6a).

During periodontitis, the MMP-8 levels detected in the vestibular periodontal sites (2755.10 pg/mL) were significantly higher than those detected in the palatine (2066.31 pg/mL, *p* < 0.001), distal (2197.26 pg/mL, *p* < 0.001), and mesial (2207.06 pg/mL, *p* < 0.001) sites (Figure 6b). The MMP-8 levels detected in the palatine sites increased during the orthodontic treatment at 3 months (2398.12 versus 1842.77, *p* = 0.001; 1935.71, *p* = and 1776.58, *p* = 0.009), 12 months (2300.27 versus 1866.06, *p* = 0.024; 1827.13, *p* = 0.014; and 1753.98, *p* = 0.011), 18 months (2182.87 versus 1787.34, *p* = 0.013; 1743.13, *p* = 0.005; and 1703.80, *p* = 0.004), and 24 months (1899.92 versus 1616.67, *p* = 0.009; 1676.03, *p* = 0.013; and 1566.67, *p* = 0.003) when compared with those detected in the mesial, distal, and vestibular periodontal sites, respectively (Figure 6b).

**Figure 6 ijms-24-04807-f006:**
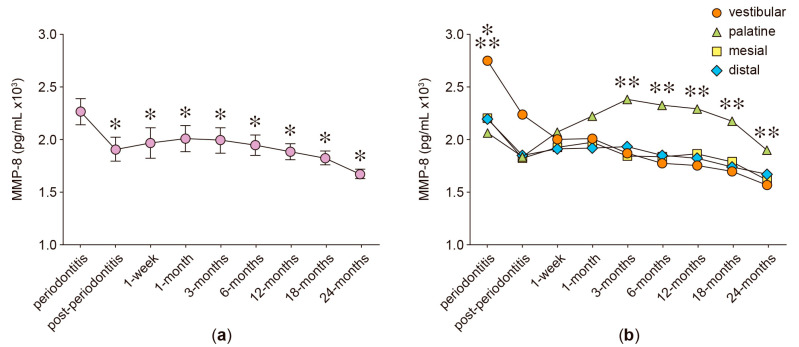
MMP-8 levels. (**a**) MMP-8 levels detected in the gingival crevicular fluid of periodontally compromised anterior teeth before periodontitis treatment (periodontitis), after periodontitis treatment (post-periodontitis), and during 2 years of patient-specific orthodontic treatment (1 week to 24 months), using intermittent low-intensity orthodontic forces. Data are represented as pg/mL and shown as mean ± SD. (**b**) MMP-8 levels detected in each periodontal site, analyzed at the same time-points described in (**a**). * *p* < 0.05 periodontitis compared with post-periodontitis and all the time-points of orthodontic treatment. ** *p* < 0.05 at 3-month to 24-month examinations, comparing the palatine site with the other periodontal sites. *** *p* < 0.05 at periodontitis examination, comparing the vestibular site with the other periodontal sites.

## 3. Discussion

Osteoimmunology is a discipline that studies the interplay between the immune and skeletal systems [16]. This discipline establishes a molecularly regulated cellular coupling between immunity and bone metabolism, where the immuno-inflammatory response can modulate key stages of bone resorption, such as the differentiation and activation of osteoclasts [17,18]. In this context, orthodontic forces trigger an aseptic immuno-inflammatory process within the periodontal tissues, in such a way that the tissue-resident cells and the infiltrating immune cells are able to produce pro-inflammatory and pro-destructive mediators that lead to changes in the periodontal metabolism and ultimately determine the orthodontic movements [19,20]. In the present study, the levels of different pro-inflammatory and pro-destructive mediators common between periodontitis and orthodontics were quantified, and it was shown that the levels of the RANKL, OPG, IL-6, IL-17A, and MMP-8 do not vary during the 2 years of orthodontic treatment. Interestingly, the RANKL/OPG ratio and the MMP-8 levels were lower at all the analyzed time-points of the orthodontic treatment compared to those detected during periodontitis. These results allow us to suggest that the used orthodontic treatment method ensures dental movements without deleteriously affecting the periodontal tissues, at least molecularly, and therefore, it would not trigger any additional periodontal insertion loss in the treated teeth.

The interaction between RANKL and its specific receptor RANK determines both the differentiation of osteoclasts from monocyte precursors and the activation of mature osteoclasts [21]. In this context, OPG competes with RANK to bind to RANKL and thus, inhibits the osteoclast activity [22]. Therefore, an increase in the RANKL/OPG ratio is associated with osteoclast-mediated bone resorption [23], which could be due to an increase in the RANKL levels, a decrease in the OPG levels, or both simultaneously. This increment in the RANKL/OPG ratio has also been associated with the onset and severity of periodontitis [23]. Similarly, an increase in the RANKL/OPG ratio has also been described during orthodontic movements [14]. In the present study, the RANKL/OPG ratio remained stable during the 2 years of orthodontic treatment, reflecting its role during the bone changes associated with dental movements; however, these levels were significantly lower than those produced during periodontitis. Interestingly, before the beginning of the orthodontic treatment, the protruded teeth exhibited high levels of RANKL in the GCF samples obtained from the vestibular periodontal site. However, when the orthodontic forces were applied, the vestibular levels of RANKL decreased and the palatine levels increased; thus, reflecting the main force component of the orthodontics movement directed towards the palatine surface.

Recently, three systematic reviews revealed that conventional orthodontic treatment could affect the periodontal status [20,24,25]. In particular, it was established that an increase in periodontal parameters occurs during orthodontic treatment, affecting the composition and accumulation of the subgingival microbiota and generating an increase in gingival inflammation and bleeding on probing [20,24,25]. In this context, it has been shown that IL-6 levels increase during orthodontic treatment, a pro-inflammatory cytokine characteristic of the Th17-pattern of the immune response, so this increase can explain, at least in part, the described inflammatory changes [26,27]. The Th17-type of cytokines, IL-6 and IL-17A, have been implicated in diverse bone resorptive conditions and diseases [28]. Indeed, the orthodontic forces induce the activation of periodontal Th lymphocytes, and the Th17 lymphocytes have been directly and indirectly associated with the production of RANKL [14,27]. On the one hand, Th17 lymphocytes have the ability to induce the differentiation and activation of osteoclasts through the direct production of RANKL [28,29,30]. On the other hand, IL-6 and IL-17A can exert osteoclastogenic activity by inducing the expression of RANKL in osteoblasts and fibroblasts [31,32,33]. Moreover, IL-17A facilitates local inflammation by promoting the recruitment and activity of monocytes and macrophages, which lead to an abundance of other pro-inflammatory cytokines, such as IL-1β and TNF-α, which in turn induce an increase in the expression of RANKL in Th17 lymphocytes [34,35]. Thus, the activity of Th17 lymphocytes could be associated with the production of IL-6 and IL-17A detected during the orthodontics movements. However, in the present study, the IL-6 and IL-17A production during the 2 years of orthodontic treatment was lower than those detected during previous periodontitis.

MMPs play a key role during the homeostatic remodeling and the pathological degradation of the extracellular matrix components of the periodontal tissues [36]. Whereas the activity of most MMPs is low in a healthy periodontium, significantly greater activity of MMPs is detected during periodontal inflammation [36]. During orthodontic movements, the MMP levels detected in the GCF reach their maximum in an average of 1 to 2 days after the application of the orthodontic forces and return to their initial values approximately 1 week after treatment [37]. Similarly, the MMP-8 levels are increased in GCF during periodontitis and decrease after successful non-surgical periodontal therapy [38,39], which agrees with the results obtained in this study, when higher levels of MMP-8 were observed during periodontitis, and these levels decreased after the periodontal disease was resolved.

In this line, the levels of MMP-8 at the periodontitis examination were higher in the vestibular periodontal site in comparison to the other sites, which is consistent with the protruded pathological position of the affected teeth. After initiating the orthodontic treatment, the vestibular levels of MMP-8 decreased and the MMP-8 levels in the palatine periodontal site increased in comparison to the other sites, reflecting the greater compression in the palatine periodontium during the orthodontic treatment. In general terms, the average levels of MMP-8 gradually decreased during the 2 years of orthodontic treatment, at all the analyzed times being lower than those detected during periodontitis. Certainly, the type of orthodontic treatment used could contribute to the differences found between the present study using intermittent low-intensity orthodontic forces and the previous reports that showed significantly elevated levels of MMP-8 after applying conventional orthodontic forces [40,41,42]. In addition, differences can be explained by the distinct study methodology used. In particular, the present work analyzed patients in orthodontic treatment after the periodontitis was resolved and who were undergoing supportive periodontal therapy. In contrast, in previous reports, healthy subjects without antecedents of periodontal disease were studied [40,41,42].

Herein, we analyzed the molecular feasibility of performing an orthodontic treatment in a secure way for patients already treated for periodontitis and with an alveolar bone remnant inferior to 50%. The main limitation of this study is that we only analyzed five molecular mediators involved in the changes that occur in periodontal tissues during periodontitis and orthodontic treatment. Clearly, analyzing five molecular mediators is an underrepresentation of the periodontal immuno-inflammatory response that determines these changes. Even so, we consider that the molecular mediators that we studied are representative of the pathophysiological phenomena of interest: periodontal inflammation, soft tissue degradation, and tooth-supporting alveolar bone loss. On the other hand, this study involved a before–after analysis of these molecular mediators, and the previous periodontitis condition was used for comparisons. By treatment structure, the present study could have been raised as a controlled clinical trial with a parallel, replacement, cross-over, or factorial design. However, the ethical concerns involved in studying a control group without orthodontic treatment are important. The same occurs with a group of patients undergoing conventional orthodontic treatment; in particular, if the background that justifies the present study indicates that an orthodontic treatment based on controlled low-intensity orthodontic forces allows the achievement of clinical success [3,43].

Based on the findings of this study, we can establish that the orthodontic treatment involving intermittent low-intensity forces does not generate molecular changes related to the destruction of teeth-supporting periodontal tissues. Thus, this is a safe and recommended therapeutic strategy to correct the position of the protruded anterior teeth as a consequence of reduced periodontal support. It is clear that the expected therapeutic success also depends on the adequate control of dental biofilm, whose disruption allows the avoidance of quantitative and qualitative changes in its bacterial composition frequently observed in orthodontic patients [7,8]. Nowadays, different strategies are aimed at facilitating daily domiciliary patient-administered mechanical plaque control. Powered toothbrushes, such as rotating/oscillating and side-to-side sonic toothbrushes, have been shown to be effective in maintaining periodontal health [44]. In addition, they are more predictable than manual toothbrushes and easier for patients to learn to use [45]. As a complement, the use of probiotics, paraprobiotics, or postbiotics is also a viable alternative aimed at ensuring oral homeostasis during orthodontic treatment. Probiotics have emerged as an attractive alternative for the treatment of periodontitis and furthermore, they do not present side effects such as those produced by the prolonged use of chlorhexidine [46]. Similarly, the use of paraprobiotics-based products during periodontitis treatment has resulted in a significant improvement in periodontal parameters and a decrease in the percentage of periodontopathogenic bacteria in the affected periodontal sites [47]. In the case of postbiotics, corresponding to microbial fermentation products with antioxidant activity, clinical benefits have also been observed during the treatment of periodontitis [48]; however, in all these cases, more studies are necessary to ratify their beneficial activity in the control of periodontitis. Taken together, these novel therapeutic strategies could help the patient achieve adequate control of the dental biofilm and would favor the maintenance of a healthy oral environment that ensures a successful orthodontic treatment.

In summary, the levels of RANKL, OPG, IL-6, IL-17A, and MMP-8 analyzed in the present study did not vary during the 2 years of orthodontic treatment. Moreover, at all the analyzed time-points of the orthodontic treatment, the RANKL/OPG ratio and MMP-8 levels, which are directly associated with alveolar bone resorption and soft tissue degradation, respectively, were maintained below those detected during previous periodontitis. Controlled low-intensity orthodontic forces have been previously associated with clinical periodontal stability and the absence of an increased progression of periodontitis [3,43], including in patients undergoing periodontal regeneration [6]. The findings of the present study support these clinical findings at a molecular level by analyzing biomarkers involved in inflammation, soft tissue degradation, and alveolar bone resorption.

## 4. Materials and Methods

### 4.1. Study Population

Forty-two periodontitis-affected patients with protruded anterior teeth were treated at the Orthodontics–Periodontics Clinic belonging to the Dentistry Faculty, Universidad de Chile. Adults were treated for stages III or IV periodontitis and underwent supportive periodontal therapy for 6 months, with at least 2 periodontal reevaluations. Then, patients were orthodontically treated for 2 years to correct the pathological migration of the teeth of group II. Molecular analysis was performed in each control session, as described below.

### 4.2. Selection Criteria

The criteria for patient selection were as follows: a minimum of 14 natural teeth, excluding third molars and including at least 6 posterior teeth, no previous periodontal therapy, absence of relevant systemic diseases, and no intake of antibiotics or non-steroid anti-inflammatory drugs in the last 6 months. All the selected patients underwent a clinical periodontal examination by a single calibrated examiner (AR), who registered the periodontal parameters PD, CAL loss, and dichotomous measurements of BP and BOP at six periodontal sites in all teeth, excluding third molars. The diagnosis of periodontitis was established by having at least five teeth with PD ≥ 6 mm and CAL ≥ 5 mm, together with generalized radiographic bone loss (>30% of the periodontal sites). Patients had occlusal stability and periodontally compromised teeth belonging to group II, with radiographically assessed marginal bone loss >50%. Pregnant or lactating women and individuals using removable prostheses were not selected.

### 4.3. Periodontal Treatment and Supportive Periodontal Therapy

Non-surgical periodontal treatment was performed under troncular anesthesia, scheduled within two weeks. During periodontal treatment, patients rinsed twice a day with 15 mL of 0.12% chlorhexidine gluconate solution for 30 s. Oral hygiene instruction was performed throughout full treatment, and a rigorous, supportive periodontal therapy was performed with periodontal maintenance reevaluations every 3 months. When an effective full-mouth plaque control was assured, with BP ≤ 10% and BOP ≤ 10%, patients underwent orthodontic treatment.

### 4.4. Orthodontic Treatment

The orthodontic treatment was performed by a single orthodontist (CN), and the patients were referred to controls every four weeks. First, 0.022-inch pre-adjusted brackets (MBT™ brackets, Balance GAC^®^, Milford, DE, USA) and tubes (Ovation GAC^®^, Milford, DE, USA) were placed in the maxillar and mandibular arches. A patient-specific strategy for the cementation of brackets was used, which consisted of placing them on the vestibular surfaces of the dental crowns according to the remaining bone level of each tooth instead of the center of the clinical crowns. The first archwire installed was a 0.0155-inch three-strand braided stainless steel wire, compensating on each tooth according to the heights defined by the bracket cementation. Approximately 4 to 6 months later, once the rotations were corrected, a 0.014-inch stainless steel wire arch was placed, with which the active intrusion phase was initiated by using intrusion bends no greater than 1 mm in each of the extruded teeth. When the leveling and alignment of the maxillary and mandibular arches were completed, fixed lingual retainers of 0.0175-inch of three-strand braided stainless steel wire were installed for retention in both maxillary and mandibular incisors. Then, 0.016-inch stainless steel archwires with a single closing loop distal were installed for closing scattered spaces. After 18 months of treatment, an upper arch 0.016-inch and lower arch 0.014-inch stainless steel archwires were installed, to start the fine-adjustment phase of treatment.

### 4.5. Clinical Evaluation

The patients were clinically evaluated for 2 years by a single periodontist (AR). Standardized clinical measurements at all time-points were performed at all periodontal sites around each tooth. The reproducibility of the examiner was determined before the beginning of the study. For this, 10 patients with stage III or IV periodontitis, similar to the patients included in this study but not considered for this work, were enrolled. The PD, CAL, BP, and BOP parameters were recorded from the whole mouth within a 7 days interval. The intra-examiner reproducibility for the NIC was calculated. The examination was considered reproducible after reaching a record of coincidence of ~1 mm in >90% of the repeated measurements.

### 4.6. Time-Points of GCF Sampling

The following time-points were established as a reference for the study: T-1: pre-periodontal examination, 1 week prior to periodontal treatment; T0: pre-orthodontic examination, at 6 months of successful supportive periodontal therapy; T1: 1 week after initiating the orthodontic treatment; T2: 1 month after initiating the orthodontic treatment; T3: 3 months after initiating the orthodontic treatment; T4: 6 months after initiating the orthodontic treatment; T5: 12 months after initiating the orthodontic treatment; T6: 18 months after initiating the orthodontic treatment; and T7: 24 months after initiating the orthodontic treatment.

### 4.7. GCF Sampling

Four periodontal sites were examined per anterior affected tooth: vestibular, palatine, mesial, and distal. Teeth were isolated in a relative manner using cotton rolls, and the bacterial supragingival plaque was removed using a ¾ curette (Hu Friedy, Gracey, IL, USA), without touching the marginal gingiva. The periodontal site was then dried gently with an air syringe, and the GCF was collected using paper strips (ProFlow™, Amityville, NY, USA), according to the manufacturer’s instructions. The paper strips were softly placed into the gingivodental crevice until reaching minimal resistance and kept in place for 30 s. Strips contaminated by saliva or blood were excluded. After GCF collection, strips were placed in sterile vials containing 100 μL of 0.05% Tween-20 in phosphate-buffered saline and centrifuged for 10 min at 10,000× *g* at 4 °C. The elution procedure was repeated twice, and samples were stored at −20 °C until further analysis.

### 4.8. Production of RANKL, OPG, IL-6, IL-17A, and MMP-8

The secreted levels of RANKL, OPG, IL-6, IL-17A, and MMP-8 were determined using specific ELISA kits (Quantikine^®^ or DuoSet^®^ ELISA Kits, R&D Systems Inc., Minneapolis, MN, USA), following the manufacturer’s recommendations. Data were obtained using an automated plate spectrophotometer (Synergy™ HT, Bio-Tek Instrument Inc., Winooski, VT, USA).

### 4.9. Statistical Analysis

The sample size was determined using the G*Power 3.1 software (Heinrich Heine Universität Düsseldorf, Germany), with a 5% significance level (α = 0.05) for a 2-tailed test, an 80% statistical power (1 − β = 0.80), and a dropout of 25%, based on one of our previous studies with follow-up of patients for at least 2 years after periodontitis treatment [49]. Considering RANKL as a strong variable, the sample size necessary to detect concentration differences of at least 71 pg/mL was determined, with a standard deviation of 103.4 [50].

The ELISA results were calculated using a logistic equation of 4 parameters and expressed as mean ± standard deviation. Data were statistically analyzed (SPSS 22.0, IBM Corp., Armonk, NY, USA), and the normality of data distribution was determined with the Kolmogorov–Smirnov test. The differences were analyzed using the ANOVA and post hoc Bonferroni tests. Statistical significance was established when α < 0.05.

## 5. Conclusions

In the present study, the levels of RANKL, OPG, IL-6, IL-17A, and MMP-8 did not vary during the 2 years of orthodontic treatment involving intermittent low-intensity forces. In particular, the RANKL/OPG ratio and MMP-8 levels, which are directly associated with alveolar bone resorption and soft tissue degradation, respectively, were maintained below those detected during previous periodontitis. Therefore, the results of the present study could allow us to suggest that it is possible to perform an orthodontic treatment safely in patients affected with pathological migration of anterior teeth as a consequence of periodontitis-associated reduced periodontal support. The recommended protocol must involve a rigorous periodontal maintenance program and orthodontic treatment with forces of low intensities in an adequate manner for each particular case.

## Figures and Tables

**Table 1 ijms-24-04807-t001:** Clinical characteristics of the studied subjects.

	Pre-Orthodontic Examination	Orthodontic Treatment	
		12 Months	24 Months
Age (years; mean and range)	42 (25–55)		
Women (number and percentage)	26 (61.91%)		
Smoking subjects (number of subjects)	2	0	0
Clinical attachment level full mouth (mm; mean ± SD)	5.13 ± 1.06	5.25 ± 0.81	5.29 ± 0.77
Clinical attachment level in the anterior teeth (mm; mean ± SD)	5.89 ± 1.37	5.93 ± 0.93	5.91 ± 0.88
Probing depths full mouth (mm; mean ± SD)	2.26 ± 0.42	2.25 ± 0.40	2.21 ± 0.39
Probing depths in the anterior teeth (mm; mean ± SD)	2.83 ± 0.47	2.77 ± 0.50	2.75 ± 0.48
Supragingival bacterial plaque full mouth (percentage)	6	5	5
Supragingival bacterial plaque in the anterior teeth (percentage)	4	2	3
Bleeding on probing full mouth (percentage)	4	4	4
Bleeding on probing in the anterior teeth (percentage)	0	0	0

## Data Availability

Not applicable.

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
