# Peer review of "Levels of Pro-Inflammatory and Bone-Resorptive Mediators in Periodontally Compromised Patients under Orthodontic Treatment Involving Intermittent Forces of Low Intensities"

_ijms, 2023, doi:10.3390/ijms24054807_

Round 1

Reviewer 1 Report

Please provide the full form of all abbreviated forms at their first time mentions (for example, line 65, PD, CAL, BP, and BOP).

Please include a section naming limitations of the study.

Please include a concluding sentence based on the research findings in the conclusion section.

If possible, include a graphical abstract.

Author Response

Reviewer #1.

Comment #1: Please provide the full form of all abbreviated forms at their first time mentions (for example, line 65, PD, CAL, BP, and BOP).

Response #1: As requested, the full form of all the abbreviations has been provided the first time they were mentioned in the manuscript, including the abbreviations of the clinical parameters characteristic of periodontitis: PD, CAL, BP, and BOP.

Comment #2: Please include a section naming limitations of the study.

Response #2: As requested by the Reviewer, we have added a new paragraph in the discussion section of the present version of the manuscript detailing the main limitations of our study. In particular, we emphasize that only five molecular mediators were analyzed to assess the periodontal tissue stability during the 2-years orthodontic treatment. It is clear that analyzing five molecular mediators is an underrepresentation of the periodontal response that determines the changes in periodontal tissues that occur during periodontitis and orthodontic treatment. Even so, we consider that the chosen molecular markers are representative of the following tissular events described as critical in the periodontium: inflammation, soft tissue degradation, and alveolar bone resorption.

Comment #3: Please include a concluding sentence based on the research findings in the conclusion section.

Response #3: Following the Reviewer´s recommendation, two concluding sentences based on our research findings have been added in the Conclusion section of the present version of the manuscript.

Comment #4: If possible, include a graphical abstract.

Response #4: According to this recommendation, a graphical abstract has been included in this revised manuscript version.

Reviewer 2 Report

This manuscript entitled “Levels of pro-inflammatory and bone-resorptive mediators in periodontally compromised patients under orthodontic treatment involving intermittent forces of low intensities” has been reviewed thoroughly. In this study, the authors showed that the patient-specific orthodontic treatment was well tolerated by periodontally compromised teeth with pathological migration.

1. It would be better to add some content about the number of patients receiving orthodontic treatment in Result 2.1.

2. According to the number of patients mentioned in Study population, the percentage of women in Table 1 was irrational. There may be some errors in one of the two.

3. The site from which the GCF sampled in Figure “a” came from and whether the data in Figure “a” and Figure “b” was duplicated?

4. Line 143, changing “no differences” to “no significant differences” would be more accurate.

Author Response

Reviewer #2.

This manuscript entitled “Levels of pro-inflammatory and bone-resorptive mediators in periodontally compromised patients under orthodontic treatment involving intermittent forces of low intensities” has been reviewed thoroughly. In this study, the authors showed that the patient-specific orthodontic treatment was well tolerated by periodontally compromised teeth with pathological migration.

Comment #1: It would be better to add some content about the number of patients receiving orthodontic treatment in Result 2.1.

Response #1: Following this recommendation, the number of patients who were treated for periodontitis and then received two years of orthodontic treatment has been added in the manuscript's first paragraph of the results section.

Comment #2: According to the number of patients mentioned in Study population, the percentage of women in Table 1 was irrational. There may be some errors in one of the two.

Response #2: As the Reviewer pointed out, we made a mistake when displaying the percentage of analyzed women, due to a decimal approximation. In order to clarify this information, the number of women participating in the study has been detailed as the absolute number and percentage in Table 1 of the manuscript.

Comment #3: The site from which the GCF sampled in Figure “a” came from and whether the data in Figure “a” and Figure “b” was duplicated?

Response #3: We appreciate the Reviewer noticing that the information related to the Figures may lead to some confusion. In particular, we would want to clarify at this point that the information contained in panels “a” and “b” of each Figure is different. In each Figure, panel “a” shows the total concentration (or ratio of the RANKL and OPG concentration, in the case of Figure 3) of each molecular mediator, corresponding to all the sampled periodontal sites in each tooth. Instead, panel “b” shows the concentration of each molecular mediator per periodontal site. Thus, panel “a” allows the comparison of molecular levels between the different evaluation time-points of the orthodontic treatment and when the patient was affected with periodontitis. In contrast, panel “b” allows the comparison of the variations of the molecular mediator between each of the four periodontal sites.

To better clarify this issue, a new section has been added to the manuscript (section 2.2. Molecular evaluation) describing the differences between the panels in each Figure.

Comment #4: Line 143, changing “no differences” to “no significant differences” would be more accurate.

Response #4: The change has been done as requested.

Reviewer 3 Report

Manuscript of considerable interest for the dental sector, requires a MAJOR REVISION

Abstract: to add the results, highlighting them more

Keywords: add specifics

Introduction; how does the oral microbiota change in the orthodontic patient? add the study by Prof. Sfondrini et al.

Very confusing results, reorganize them to make it easier for the reader

Discussion: add as future goals the use of probiotics, paraprobiotics and postbiotics to regulate oral cavity homeostasis and use an electric and/or sonic toothbrush, as studied by the research group of prof Scribante et al.

Materials and methods, how was the sample size calculated?

Conclusions: add proactive action.

Bibliography; add references required

Author Response

Reviewer #3:

Manuscript of considerable interest for the dental sector, requires a Major Revision.

Comment #1: Abstract: to add the results, highlighting them more.

Response #1: Following the Reviewer´s recommendation, we have better highlighted the findings of our study in the present version of the Abstract, while considering the recommended word limit for this section.

Comment #2: Keywords: add specifics.

Response #2: In the revised version of our manuscript, we have added more specific keywords.

Comment #3: Introduction; how does the oral microbiota change in the orthodontic patient? add the study by Prof. Sfondrini et al.

Response #3: Following the Reviewer´s suggestion, a new paragraph has been added to the Introduction section of the manuscript. In this new paragraph, we discuss the quantitative and qualitative changes in the oral biofilm observed in orthodontically treated patients, highlighting the possible appearance of bacterial species related to gingivitis and periodontitis etiology. In addition, we emphasize that the periodic follow-up of the patient and the adequate control of the patient´s dental biofilm ensure that these microbial changes do not translate into deleterious effects over the periodontal tissues. In support of this newly added paragraph, the reference mentioned by the Reviewer, as well as other references related to the same topic, have been added to the References section of the present version manuscript.

Comment #4: Very confusing results, reorganize them to make it easier for the reader.

Response #4: Following the Reviewer's recommendation, the Results section of the manuscript has been completely modified. In this revised version, the sentences that present the results have been rewritten, presenting the information more directly. In addition, section 2.2 has been added, which explains more clearly how the results are shown in the figures. In line with these changes, the figure legends have also been rewritten, and the text in the figures has been changed to match the legends.

We hope that all these changes will allow potential readers of our manuscript to better understand the main findings of the study.

Comment #5: Discussion: add as future goals the use of probiotics, paraprobiotics and postbiotics to regulate oral cavity homeostasis and use an electric and/or sonic toothbrush, as studied by the research group of prof Scribante et al.

Response #5: Following the Reviewer's suggestion, a new paragraph has been added to the Discussion section of the present version of the manuscript. In this new paragraph, we discuss the use of probiotics, paraprobiotics, and postbiotics, as well as powered toothbrushes, as adjuvants to ensure the maintenance of oral homeostasis during orthodontic treatment. According to this, five new references have been added.

Comment #6: Materials and methods, how was the sample size calculated?

Response #6: In accordance with the Reviewer´s recommendation, a new paragraph detailing how the sample size was calculated has been added to the Statistical analysis section of the present version of the manuscript. According to this, two new references have been added.

Comment #7: Conclusions: add proactive action.

Response #7: Following the Reviewer´s suggestion, two new concluding sentences based on the main findings of our study have been added in the Conclusion section of the manuscript.

Comment #8: Bibliography; add references required.

Response #8: Accordingly, all the references that support the new sentences and paragraphs that were added have been included in the References section of the manuscript.

Round 2

Reviewer 3 Report

The manuscript has been revised in every part. It can be published